# Improved Rate of Negative Margins for Inflammatory Breast Cancer Using Intraoperative Frozen Section Analysis

**DOI:** 10.3390/cancers15184597

**Published:** 2023-09-16

**Authors:** Joshua Kong, Sudeshna Bandyopadhyay, Wei Chen, Faisal Al-Mufarrej, Lydia Choi, Mary A. Kosir

**Affiliations:** 1Department of Surgery, Wayne State University, 4160 John R, Suite 400, Detroit, MI 48201, USA; 2Karmanos Cancer Institute, 4100 John R, Detroit, MI 48201, USAchenw@karmanos.org (W.C.);; 3Department of Pathology, Wayne State University, 540 E. Canfield, Ste. 9374, Detroit, MI 48201, USA; 4Division of Plastic Surgery, Department of Surgery, Wayne State University, 4160 John R, Suite 400, Detroit, MI 48201, USA

**Keywords:** inflammatory breast cancer, intraoperative frozen section analysis, negative margin

## Abstract

**Simple Summary:**

Achieving negative surgical margins following modified radical mastectomy for the treatment of inflammatory breast cancer is critical to the survival of inflammatory breast cancer patients. The current surgical technique is reported to underestimate skin tumor infiltration by up to 60%. The aim of our prospective study was to assess the potential benefit of improving the rate of negative surgical margins for inflammatory breast cancer using intraoperative frozen section analysis.

**Abstract:**

Background: Inflammatory breast cancer (IBC) is a rare and aggressive form of breast cancer with a poor survival rate. Modified radical mastectomy (MRM) with negative pathologic margins is critical for improved survival. We aim to study the potential benefit of intraoperative frozen section analysis (FSA) to improve disease-free margins. Methods: This prospective, monocentric study included 19 patients who underwent MRM for IBC. For each patient, a 2 mm continuous skin edge was sent for FSA to guide further resection. The rate of tumor-free margins and the concurrence between the FSA and permanent pathological results were analyzed. Results: Overall, 15 of the 19 patients achieved negative margins, including four patients who would have had positive margins without FSA. The odds ratio of achieving a negative final margin with FSA was infinity (*p* = 0.031), and there was a strong agreement between the FSA and permanent pathological results (Kappa—0.83; *p* < 0.0001). Conclusions: The FSA technique decreased the number of positive margins in IBC patients undergoing MRM, thereby potentially reducing the need for re-operation, allowing immediate wound closure, and preventing delays in the administration of adjuvant radiation therapy. More extensive trials are warranted to establish the use of intraoperative FSA in IBC treatment.

## 1. Introduction

Inflammatory breast cancer (IBC) is a rare and aggressive cancer, representing 2–4% of all breast cancers but contributing to 10% of cancer-related mortalities [1,2,3]. Significant improvements have been reported for patients with IBC using a trimodal regimen of neoadjuvant chemotherapy, followed by modified radical mastectomy (MRM) and postoperative radiation therapy [1,4,5,6,7,8,9]. Surgeons play a critical role in achieving better outcomes in that obtaining a negative pathological margin with surgical resection significantly enhances survival outcomes [10,11,12,13]. Achieving a negative margin during the initial operation prevents the need for re-excision, which delays the administration of subsequent radiation therapy and increases problems with wound complications. This report focuses on a technique for optimizing disease-free margins in a clinical setting, where clinical surgical assessments often underestimate cutaneous infiltration by as much as 60% due to dermal lymphatic infiltration by the tumor in patients with erythema and peau d’orange associated with IBC [14,15]. This feasibility study was designed to assess the efficacy of intraoperative frozen section analysis (FSA) in guiding surgeons to achieve negative margins during the index operation. Although FSA has often been reported as an adjunct associated with lumpectomy, it has not been described for IBC [16,17].

## 2. Design and Methods

**Study Design and Patient Selection Criteria.** During a 12-year interval ending in 2020, 19 patients with non-metastatic IBC were prospectively analyzed. All 19 patients had good clinical response to neoadjuvant chemotherapy and were deemed to be good candidates for MRM as judged by the response of decreased skin edema, redness, palpable lymph nodes, and peau d’orange. None of the patients required preoperative radiation therapy. Their operative consents explained that this is a prospective method to assess margins intraoperatively using standard frozen section analysis, where their results may be used for clinical assessment of this modified approach, and that there may be subsequent complications using their results. The patients were assured that no personal identification factors would be released.

**Intervention.** The incision for the MRM was determined by the surgeon to encompass any remaining abnormal skin as well as to facilitate closure. After the incision was made, a 2 mm wide edge was excised as a continuous piece from the skin ellipse, which was part of the planned specimen, oriented and sent to the pathologist for FSA (Figure 1). In the meantime, the MRM proceeded. The pathologist cut the specimen into 4 or 8 pieces and sectioned the outer edges for the frozen section analysis. While the operation was continuing, the results were verbally reported to the operation room. If all results were negative on the first FSA, no further patient skin would be submitted for FSA (Figure 2). The pathologist submitted all frozen section specimens for permanent section analysis. When local intradermal lymphatic tumor emboli were reported by the pathologist, the location of the area directed the surgeon to resect an additional patient skin edge which was also oriented and sent fresh to the pathologist for FSA while still in the operating room. Occasionally, there would be multiple areas of cancer cells detected by means of FSA. This process was repeated until a negative FSA was achieved, or the surgeon determined that no additional excision of patient skin could be performed and still achieve adequate wound closure. In this circumstance, the final patient skin edge biopsy was sent for permanent analysis rather than FSA. By resecting additional patient skin edges and repeating the FSA, negative frozen section results were ultimately reported in most patients (Figure 2). In three patients where additional skin resection was deemed not desirable by the surgeon (overlying clavicle or crossing midline), an intraoperative decision was made to stop obtaining FSA sections for the microscopic foci detected in minute areas, with the remaining patient skin edges being negative. All frozen section activities occurred while the MRM was being performed to avoid extra time in the operating room. Following resection by MRM, most patients underwent immediate primary closure. For two patients, closure was preoperatively planned to use either latissimus dorsi flap or rhomboid flap and was performed by the plastic surgeon without delay.

**Statistical Method**. This was a single-institution retrospective study with self-control. Each patient had one or more consecutive tissue specimens from the circumferential excisional biopsy. The primary endpoint was the association between the method, FSA vs. permanent pathology (non-FSA), and the proportion of achieving negative final margin. A secondary endpoint was the agreement between the FSA and the permanent pathological results. The analyses of the FSA were compared to the permanent section analyses to determine the accuracy of the FSA. Descriptive statistics were provided with 95% confidence intervals. A swimmer plot was provided to outline the results of the FSA with positive and negative outcomes. The agreement between the FSA and permanent pathological results was summarized at the specimen level and evaluated using Cohen’s Kappa statistic. The Cohen’s Kappa result should be treated descriptively as it assumes independence of tissue specimens, ignoring the correlation among consecutive pairs from a small proportion of patients in the specimen-level data analysis. As our primary objective, the analysis of using FSA in achieving a higher proportion of negative final margin compared to the non-FSA method was summarized from the patient-level data. We compared the proportion of negative margins using the final permanent pathology (the gold standard for the FSA method) to the proportion of negative margins using the first permanent pathology (patient’s self-control as if the non-FSA method was used). The primary endpoint was tested using the one-sided mid-p exact McNemar test for binary outcomes from paired small samples [18]. The analyses were performed using the R statistical software version 4.1.0 (R Foundation for statistical Computing, Vienna, Austria).

## 3. Results

Patient Characteristics. The 19 patients prospectively included in this study were all female, with an average age of 54.4 years (range 38–78 years). This included 12 (66.7%) African Americans, six (31.6%) Caucasians, and one (5.3%) Hispanic (Table 1). The MRM specimens obtained from the 19 patients, as summarized in Table 1, demonstrated a complete response in one patient, residual ductal cancer in situ in two patients, and residual disease in 16 patients ranging from T1 to T4 disease. Five patients had no residual nodal disease and eleven patients had N2 or N3 disease. Tumor markers were triple-negative and ER+/HER2neu-negative disease accounted for most cases, with six patients each, while the remaining patients had triple-positive, ER+/HER2neu+, and ER-/HER2neu+ disease.

Sequence of FSA-Guided Technique. Figure 2 broadly shows the algorithm for the intraoperative FSA. For the first skin sample obtained for FSA, 12 out of 19 patients were negative for cancer cells and seven patients were positive. Of the 12 negative FSA samples, 11 samples were confirmed by the permanent sections to be true negatives, while one was a false negative. In Figure 3, the 12 samples with negative initial intraoperative frozen sections are shown in the swimmer plot with an aqua bar, with final permanent pathological results shown as negative ( ● ) or positive ( ■ ) at the end of the aqua bar. The seven cases with positive margins based on the FSA are shown with an orange bar.

Of the seven patients with initial positive FSA results, between one and five additional resections were collected per patient. Using the FSA-guided technique to direct further resection, four patients achieved negative margins after subsequent FSA (Figure 2). Two of these patients (#8 and #19) had permanent sections that confirmed FSA true negative margins, while the other two (#1 and #12) showed FSA false negative margins (Figure 3). For the remaining three patients, persistent positive margins for malignancy were seen on the FSA after multiple resections of skin (Figure 2). A final section was sent for permanent pathology, with or without FSA, once it was deemed that further resection was not possible due to anatomical limitations. In two (#2 and #4) of these three patients, we were able to obtain negative margins based on the analysis of the final permanent pathological sections (Figure 3). One patient (#16) did not achieve the FSA-negative margin despite five additional resections. In summary, with the use of FSA, four additional patients out of the seven patients with an initial positive FSA, who would have otherwise had positive final margins based on only the surgeon’s intraoperative clinical assessment, achieved negative margins instead.

Agreement between FSA and Permanent Pathology Results. A total of 36 pairs of FSA and permanent sections were collected from the 19 patients, resulting in an average of two frozen sections per patient. The agreement between the FSA and the permanent pathological results was strong (Kappa = 0.83; 95% CI (0.64, 1); *p* < 0.0001) [19].

Association between the Methods (FSA vs. non-FSA) and Achieving Negative Final Margin. The advantage of intraoperative FSA in achieving a negative final margin showed significant difference. The odds ratio of achieving a negative final margin using FSA was infinity when compared to without FSA (OR = Inf; 95% CI (1.28, Inf); one-sided mid-p exact McNemar *p* = 0.031). The proportion achieving a negative margin without FSA was 11/19 (0.58; 95% CI (0.36, 0.77)), and the proportion achieving a negative margin with FSA was 15/19 (0.79; 95% CI 0.57, 0.91). The absolute risk reduction = 0.21% and the relative risk reduction = 50%, where the number needed to treat = 5 (OR = Inf; 95% CI (1.28, Inf); one-sided mid-p exact McNemar *p* = 0.031).

## 4. Discussion

Achieving a negative surgical margin following the initial procedure has multiple benefits. It allows immediate definitive wound closure, prevents additional return to the operating room for re-excision, allows earlier initiation of adjuvant radiation therapy, and avoids subjecting patients to additional psychological, physical, and financial burdens [20]. Additionally, clinical trials report that having a negative surgical margin is a critical prognostic factor for better overall survival in IBC [9,10,21]. This most likely reflects the improved survival of patients who respond well to neoadjuvant chemotherapy, as these patients are the most likely to achieve negative margins.

The current accepted strategy to achieve a negative surgical margin is to have an aggressive surgical approach [14,15,21]. In this study, we proposed an approach to achieve a higher rate of disease-free margins during IBC mastectomy following neoadjuvant chemotherapy, using elliptical skin excision with intraoperative FSA, which has been shown to have a high degree of positive correlation with final pathology. This technique potentially provides an additional tool for surgeons because the extent of disease involvement does not always correlate with clinical exams or preoperative image studies [22]. Even with magnetic resonance imaging, which can delineate thickened skin and breast cancer changes in IBC, it is often difficult to accurately determine residual cancer after neoadjuvant chemotherapy [23]. In our study, surgical resection without FSA would have underestimated the extent of disease infiltration in 36.8% of the patients. With FSA, we were able to avoid reoperation and re-excision in seven patients and preserve healthy tissue for definitive wound closure at the same time when performing the MRM.

Specific rates of concordance for detecting breast cancer tumor emboli in dermal lymphatics using FSA have not been reported in the literature. However, FSA of breast tissue, and not skin, is known to be reliable and effective in reducing the need for reoperation in breast-conserving surgeries for non-IBC, with excellent positive and negative predictive values that are >90% [16,24,25,26]. Hence, a near, but not perfect Kappa statistical analysis comparing FSA and their permanent pathological counterparts may be explained by the histopathology of IBC, which can present with scattered tumor emboli on biopsies and surgical specimens [27,28].

Primary closure was achieved after mastectomy with resection to disease-free skin under the guidance of intraoperative FSA for most of the patients in this study. In one case, the patient required five re-excisions, indicating extensive disease despite systemic neoadjuvant chemotherapy. This information helped determine that the patient’s IBC was extensive, although most of the involved skin had been resected based on the results of the FSA. Preoperatively, a complex closure with latissimus muscle flap was planned and discussed with the patient. Without that information at the index operation, a positive final margin on the permanent pathology sample would have directed a return to the operating room for more unsuccessful resections, which would have delayed the necessary radiation treatment. Such cases with extensive disease involvement and multiple skin resections may define a subgroup of patients that would benefit from immediate post-mastectomy radiation therapy rather than further resection. Newer treatment approaches are also being investigated for IBC with molecular targeting therapy, which may assist these difficult cases in the future [29].

We acknowledge the limitations of our study. First, only 19 patients were included, representing 4.7% of a single surgeon’s breast cancer patients during the 12-year period. This is a limitation inherent to the rarity of IBC, which has an incidence of 2–4% of all breast cancers [2]. A follow-up study including a larger sample size from multiple sites may test the usefulness of the technique using FSA-guided skin resection for IBC. Second, the pathobiology of IBC with increased lymphangiogenesis and tumor emboli, which spread as skip lesions with microsatellite invasion of the skin, poses as a potential challenge to the accuracy of FSA in IBC [28,30]. This may make it difficult to identify the true negative margin during operation. Third, the time for MRM is not extended by the intraoperative FSA, but it does interrupt during the communication of results and sending additional patient skin edge if positive, which uses up resources both in the operating room and in the pathology department. Fourth, the location of any microscopic positive edge may not be identified well for the radiation oncologist when planning treatment. However, through an analysis of 2 mm skin ellipses, our data show promising results and advocate for more robust studies to examine the use of intraoperative FSA as a standardized part of MRM for IBC and for optimizing resection. It was encouraging to know that in 11 of 19 patients, the first FSA was negative based on the surgeon’s judgement. This provided real-time assurance that the planned incision was adequate. However, it was also beneficial to the surgeon that with FSA, 57.1% of those cases with an initial positive margin on FSA ultimately achieved negative margins with additional patient skin edge excisions.

## 5. Conclusions

The incorporation of FSA into surgical resection for IBC led to a higher rate of disease-free skin margins after operation. It successfully prevented having to return to the operating room for re-excision, which could incur possible associated complications to our patients. FSA also identified patients with extensive microscopic local disease, for whom a more extensive resection would not be beneficial. The results of our study warrant further verification in clinical trials with larger sample sizes to compare outcomes between conventional resection and optimization of negative margins guided by FSA for the treatment of IBC.

## Figures and Tables

**Figure 1 cancers-15-04597-f001:**
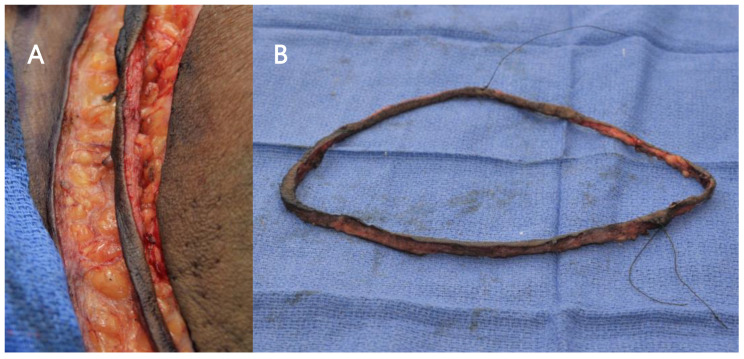
(**A**) In situ close-up view of an edge of skin ellipse that is 2 mm wide created immediately after incision. (**B**) The edge of skin ellipse oriented and sent for FSA.

**Figure 2 cancers-15-04597-f002:**
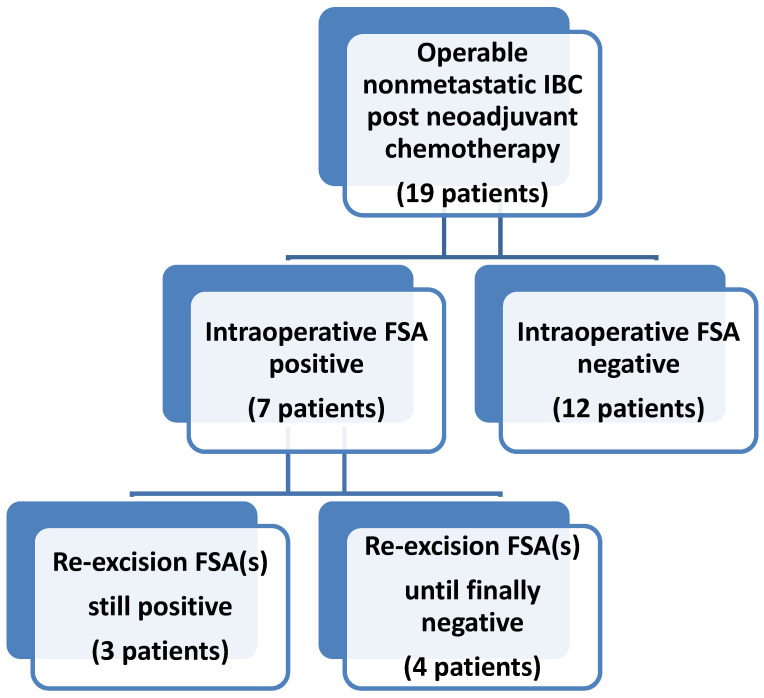
Intraoperative FSA of skin edges to assess skin margins while performing MRM for operable patients with IBC after neoadjuvant chemotherapy. If positive FSA was reported, then an additional patient skin edge is resected and assessed using FSA while still in operation, and this process is repeated multiple times, if necessary, as directed by the FSA results. Microscopic clearance of small foci of intradermal cancer cells could not be achieved for three patients while in the operating room, although the remaining skin edges are negative.

**Figure 3 cancers-15-04597-f003:**
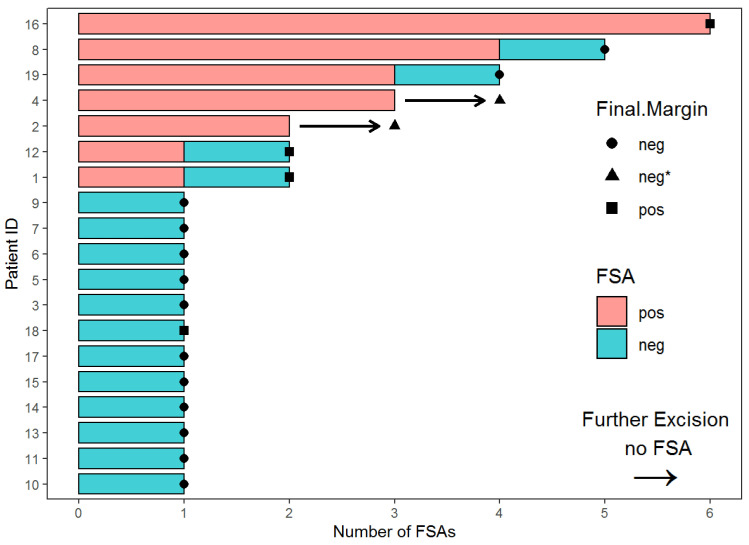
Swimmer plot of the number of FSAs per patient and results. Aqua bar depicts negative FSA, and orange bar depicts positive FSA. The arrow in place of the bar depicts the last skin resection sent for permanent section analysis rather than FSA. The final permanent section results are negative ( ● ), positive ( ■ ), and negative (▲) on permanent section without final FSA.

**Table 1 cancers-15-04597-t001:** **Patient clinical and pathological characteristics post-neoadjuvant chemotherapy**.

Variable	Number	Percentage
Age (year)	54.6 (range 38–78)	
Race		
African American	12	66.7
Caucasian	6	31.6
Hispanic	1	5.3
Residual Breast Cancer		
Complete Response	1	5.3
Tis	2	10.5
T1	2	10.5
T2	5	26.3
T3	8	42.1
T4	1	5.3
Nodal Status		
N0	5	26.3
N1	3	15.8
N2	3	15.8
N3	8	42.1
Tumor Markers		
Triple +	4	21.1
Triple −	6	31.6
ER +/PR−/HER2neu +	2	105
ER +/HER2neu −	6	31.6
ER −/HER2neu +	1	5.6

ER—estrogen receptor; HER2neu—human epidermal growth factor receptor 2.

## Data Availability

Data sharing is not applicable. No new data were created or analyzed in this study.

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
