# Peer review of "Improved Rate of Negative Margins for Inflammatory Breast Cancer Using Intraoperative Frozen Section Analysis"

_cancers, 2023, doi:10.3390/cancers15184597_

Round 1

Reviewer 1 Report

The manuscript "Improved Rates of Negative Margins for Inflammatory Breast Cancer using Intraopertive Frozen Section Analysis" demonstrates the usefulness of intraoperative frozen sections for assessing the skin resection margins.
I found the manuscript interesting and well-written.
Knowing the average number of sections required for the study and the time needed for the intraoperative diagnosis would be helpful.

Reviewer 2 Report

The topic is relevant and compliment to the AuthorsIn my opinion the work completely reflects the editorial philosophy of the Journal on the basis of the pathological and surgical evidence in the research.

Reviewer 3 Report

Dear Authors,

This is an interesting "proof of concept" study that evaluates Frozen Section Analysis during modified radical mastectomy in inflammatory breast cancer. Unfortunately, the small number of subjects and the slightly confusing study design (with the absence of a control group) limit the conclusions that can be drawn from it.

The Methodology and Results sections should be completely re-written to better clarify the design of the study, the intervention performed and the meaning of the results and statistical analysis. The Discussion section should be expanded to highlight the significant limitations of the study. The English Language should also be reviewed, as there are some minor mistakes.

Only following these changes, which represent a major revision, the article could be re-submitted for peer review and might warrant publication.

The quality of English Language was average, there are some sentences that are hard to understand and some (minor) mistakes that need to be corrected.

Round 2

Reviewer 3 Report

Thank you for revising the manuscript as requested. Your hard work and efforts are greatly appreciated, and I would be happy to accept this revised version for publication.